

# 1  Main Ethiopian Rift landslides formed in contrasting geological
# 2  settings and climatic conditions

Karel Martínek*[1,2], Kryštof Verner[2,3], Tomáš Hroch[2], Leta A. Megerssa[3,2], Veronika
Kopačková[2], David Buriánek[2], Ameha Muluneh[4], Radka Kalinová[3], Miheret Yakob[5], Muluken
Kassa[4]
*corresponding author
[1] Institute of Geology and Palaeontology, Faculty of Science, Charles University, Albertov 6,
Prague, 12843, Czech Republic (karel.j.martinek@gmail.com)
[2] Czech Geological Survey, Klárov 3, 118 21 Prague, Czech Republic
[3] Institute of Petrology and Structural Geology, Faculty of Science, Charles University, Albertov 6, Prague, 12843,
Czech Republic
[4] School of Earth Sciences, Addis Ababa University, Arat Kilo, 1176, Addis Ababa, Ethiopia
[5] Geological Survey of Ethiopia, CMC road, Bole Keb.10/Wor.6, POBox: 2302, Addis Ababa, Ethiopia
**Abstract.** The Main Ethiopian Rift (MER), where active continental rifting creates specific conditions for landslide
formation, provides a prospective area to study the influence of tectonics, lithology, geomorphology, and climate on
landslide formation. New structural and morphotectonic data from CMER and SMER support a model of
progressive change in the regional extension from NW – SE to the recent E(ENE) – W(WSW) direction driven by
the African and Somalian plates moving apart with the presumed contribution of the NNE(NE) – SSW(SW)
extension controlled by the Arabic Plate. The formation and polyphase reactivation of faults in the changing regional
stress-field significantly increase the rocks' tectonic anisotropy and the risk of slope instabilities forming.
According to geostatistical analysis landslides in the central and southern MER occur on steep slopes, almost
exclusively formed on active normal fault escarpments. Landslides are also influenced by higher annual
precipitation, precipitation seasonality, vegetation density and seasonality.
A detailed study on active rift escarpment in the Arba Minch area revealed similar affinities as in regional study of
MER. Landslides here are closely associated with steep, mostly faulted, slopes and a higher density of vegetation.
Active tectonics and seismicity are the main triggers. The Mejo area situated on the uplifting Ethiopian Plateau 60
km east of the Rift Valley shows that landslide occurrence is strongly influenced by steep erosional slopes and
deeply weathered Proterozoic metamorphic basement. Rapid headward erosion, unfavourable lithological conditions
and more intense precipitation and higher precipitation seasonality are the main triggers here.

**Keywords:**
Landslides, Main Ethiopian Rift (MER), morphotectonics, tectonics, geological setting, climate, geostatistics

## 36  1.  Introduction

Slope instabilities including mainly landslides, rockfalls and debris flows are usually influenced by key factors such
as geomorphology, bedrock lithology and rock fabric anisotropy, active tectonics and seismicity, type and grade of
weathering, climatic conditions, vegetation cover, land use and human activity. Links between these factors and the
formation of landslides and rockfalls are complex (e.g. Abebe et al., 2010; Meinhardt et al., 2015). Geomorphic
indices have been used to decipher links between landform and tectonics in several studies (Ayalew and Yamagishi,
2004; Ayalew et al., 2004). However, the influence of other factors on slope instabilities is unclear and a matter of
current debate (e.g. Asfaw, 2007; Temesgen et al., 1999; Vařilová et al., 2015; Woldearegay, 2013). In general,
ongoing discussions on the formation of slope instabilities in an active rift setting state either tectonics, climate or
anthropogenic activity as triggering factors depending on the characteristic conditions at the particular locality (e.g.
Mancini et al., 2010; Peduzzi, 2010; Wotchoko et al., 2016). Other studies also conclude that lithology and
precipitation are t he main landslide controlling factors (e.g. Kumar et al. 2019; and references therein).
Geomorphic indices, such as slope, aspect, hypsometric integral, the stream length gradient index or river incision
rates, are capable of detecting landform responses to tectonics (Ayalew and Yamagishi, 2004; Gao et al., 2013) but
studies showing slope instabilities having a direct link to active tectonics are relatively rare (Chang et al., 2018 and
references therein). Other studies also conclude that lithology and precipitation are main landslide controlling factors
(e.g. Kumar et al. 2019 and references therein).


Central and southern parts of the Main Ethiopian Rift (MER), which belong to the northern part of the East African Rift System (EARS), form a relatively narrow, slowly spreading extensional zone with a humid, strongly seasonal climate. There is a thick sequence of unconsolidated, often strongly weathered volcaniclastic deposits cropping out in grabens, on steep tectonic slopes or occasionally also on moderately elevated areas. Such a complex environment is an excellent natural laboratory to study the interplay of factors influencing various types of slope instabilities as they form in different geological and geomorphic conditions. Active extensional tectonism has a strong influence on the present-day morphology, but there are also important variations in climatic parameters (annual precipitation, seasonality); moreover, a population explosion in the last decades has led to extensive deforestation, overgrazing and dramatic changes of landcover and land use, which all may have significant importance in landslide formation (FAO 2001; Janetos and Justice, 2000; Gessesse, 2007; Gete and Hurni, 2001; Melese 2016).

This multidisciplinary study is focused on evaluating the landslide distribution in the central and southern MER. A combination of the results of geological, geohazard and structural mapping, with remotely sensed data, and climatic, vegetation and land use indicators is assessed using geostatistical methods. The discussion of the main factors influencing the formation of landslides in the regional scale in the central and southern MER and also on a detailed scale in the Mejo and Arba Minch areas in the southern part of the MER is the main focus of this study. In regional scale study the direct link to tectonics is clear, so we present large dataset of new field structural data from this area. The situation in detailed scale studies in Mejo and Arba Minch is more complex. These two areas have contrasting styles of tectonic setting and varying lithological and climatic conditions: the Mejo landslide area is more humid, located on the eastern plateau, 60 km east of the rift valley and dominated by highly weathered Proterozoic basement rocks, while the Arba Minch landslide area is situated directly on the western rift escarpment with active tectonism and seismicity, and dominated by Tertiary volcanic rocks (Fig. 1). In both areas, slope failures are closely associated with steep slopes, but these are generated by very different processes – either active rift normal faulting or deep head-ward river erosion of uplifting rift flank. The anthropogenic influence is also discussed, but only locally, because the relevant data for a thorough geostatistical evaluation are unfortunately missing.

## 2. Geological and geohazard setting
### 2.1. Geology and tectonics of the studied area

The overall geological pattern of the southern Ethiopia includes a basement formed by metamorphic rocks of the Neoproterozoic age, which have been overlaid by widespread volcanic sequences ranging from pre-rift Cenozoic volcanism to the Main Ethiopian Rift (MER) associated volcanism (Bonini et al., 2005; Hayward and Ebinger, 1996; Woldegabriel et al., 2000). The Precambrian rocks exposed in southern Ethiopia constitute the most southern part of the Arabian-Nubian Shield (ANS) which includes several terrane assemblages (for a review see Fritz et al. 2013 and references therein). The ANS is an assemblage of juvenile low-grade volcano-sedimentary rocks and associated plutons and ophiolite traces with ages between ~890 and 580 Ma (Fritz et al., 2013). The Main Ethiopian Rift (MER), is an active intra-continental rift bearing magma-dominated extension of the African (Nubian), Somalian, and Arabian lithospheric plates (e.g. Acocella, 2010; Agostini et al., 2011). Three segments of the MER reflecting temporally and spatially different stages of regional extension and volcanic activity have been defined (e.g. Hayward and Ebinger, 1996; Muluneh et al., 2014): (a) the Northern Main Ethiopian Rift (NMER), (b) the Central Main Ethiopian Rift (CMER) and (c) the Southern Main Ethiopian Rift (SMER, see Fig. 1). In the southern part of the MER, the current rate of ~E – W oriented extension between the African and Somalian plates amounts 5.2±0.9 mm/yr (Saria et al., 2014).

The volcanic activity in the studied parts of the CMER (Hossana Area) and SMER (Dilla Area) could be divided into three major episodes (Bonini et al., 2005; Corti, 2009; Hayward and Ebinger, 1996). The Eocene to Oligocene pre-rift volcanic products (~45 to 27 Ma) comprise mainly tholeite to alkaline basalt lava flows and the associated volcaniclastic deposits (Amaro-Gamo Basalts) with the presence of rhyolite ignimbrites (Shole Ignimbrites) and minor trachytes (Burianek et al., 2018; Verner et al., 2018b; Verner et al., 2018d). The Miocene syn-rift volcanic products (~22 to 8 Ma) are represented by basalts, felsic volcanites and volcaniclastic rocks (rhyolite lava, minor ignimbrites, trachyte lava flows and related pyroclastic deposits) belonging mainly to the Getra and Kele sequences including Mimo trachyte (Bonini et al., 2005; Ebinger et al., 1993; Ebinger et al., 2000). These two events were followed by a period of drastically low volcanism except for a small eruption of peralkaline pantelleritic ignimbrites intercalated with minor basaltic lava flows in the areas beyond the rift escarpments (Bonini et al., 2005; see also Fig. 4). Subsequently, the products of Pleistocene to Holocene post-rift volcanic activity (~1.6 – 0.5 Ma) are bi-modal volcanites and volcanoclastic rocks such as, for example, massive Nech-Sar basalts, rhyolites, strongly welded rhyolitic ignimbrites and other pyroclastic deposits (Ebinger et al., 1993). A typical example of post-rift volcanic



activity in the southern CMER is the lower Pleistocene formation of unconsolidated pyroclastic deposits on the rift
floor (e.g. Corbetti Volcanic System, Rapprich et al., 2014), which was consequently disturbed by tectonic
movements and erosion.
The complex fault pattern of the MER (interference of SSW(SW) -NNE(NE), N-S and WNW(W) -ESE(E) trending
faults) has been attributed to various mechanisms of contrasting hypothesis (for a review see Abate et al., 2015;
Erbello and Kidane, 2018): (a) The pure extension orthogonal to the rift; (b) a right-lateral NW – SE to the NNW –
ESE transtension continuously transferred to sinistral oblique rifting as a result of an E – W regional extension; (c) a
constant NE(ENE) – SW(WSW) trending extension; (d) constant extension in the NW – SE direction and (e)
constant E – W to ESE – WNW extension.



*Fig. 1 The Hossana and Dilla areas in the central and southern part of the Main Ethiopian Rift (MER). The location*
*of the NMER (northern MER), CMER (central MER), SMER (southern MER) and Mejo and Arba Minch case study*
*areas are also indicated. The blue lines represent major fault zones.*

**2.2.    Geohazards in the central and southern MER**
Notable geohazard features across and along the MER range from intense erosion to slope instability-related mass
wasting processes including rock falls, debris flows up to shallow and deep-seated landslides, all with immense
costs in terms of casualty and infrastructure loss (Abebe et al., 2005; Ayalew, 1999; Hearn, 2018). Landslides are
rather more common in the highlands of Ethiopia. The most affected regions are the Blue Nile Gorge (Ayalew and



Yamagishi, 2004; Gezahegn and Dessie, 1994; JICA and GSE, 2012; Tadesse, 1993), the Dessie area and the
highlands surrounding Ambassel and Woldia (Ayenew and Barbieri, 2005; Fubelli et al., 2008), the Semien
highlands, particularly western and central Tigray, the Sawla and Bonga areas of south Ethiopia (Lemessa et al.,
2000) and the MER margins of the western and eastern escarpment (Kycl et al., 2017; Rapprich and Eshetu, 2014;
Rapprich et al., 2014; Temesgen et al., 2001), the surroundings of Finchewa and the Debre Libanos and the Mugher
locality (Zvelebil et al., 2010). On the western escarpment of the MER, a vast and recurrent landslide is notable
close to the town of Debre Sina at the locality of Yizeba Weyn in central Ethiopia (Kropáček et al., 2015).
Other common geological hazards that recurrently appear in the area are ground fissures in various sectors along the
rift floor. For example, north of the Fentale area in the northern MER (Williams et al., 2004) and various localities
in the central MER segment (Asfaw, 1982; Asfaw,1998; Ayalew et al., 2004) which often transform into deep and
long gully systems (Billi and Dramis, 2003). Persistent seismic tremors, usually of lower magnitudes, are apparently
located in the entire rift floor (e.g. Wilks et al., 2017). Particular clusters and source zones have been identified in
Ethiopia those being (1) the western plateau margin, (2) the central Afar, (3) the Aisha block, (4) the Ankober area,
(5) the central Main Ethiopian Rift and (6) the South Western Main Ethiopian Rift (Ayele, 2017). Nevertheless,
historical high magnitude earthquake records have also been reported (Asfaw, 1992; Gouin, 1975; Gouin, 1979;
Wilks et al., 2017). An updated probabilistic seismic hazard analysis and zonation has since been recently carried
out with seismotectonic source zones constrained from recent studies for the Horn of Africa with reference to the
East African Rift Valley (Ayele, 2017).
In addition to the seismic tremors, volcanism is also of apparent risk. Among the recent events are the Nabro
Volcano in 2011 in the far northern part of the Afar triangle (Goitom et al., 2015) and the Debahu rifting and
volcanic dyke swarm intrusion events in 2005 (Ayele et al., 2007; Ayele et al., 2009). These two events each
triggered major alarms significant enough to warrant flight diversions (in the case of the Nabro volcano) across the
region and the temporary displacement of local people (e.g. Goitom et al., 2015).
### 3.  Methods
Field geological, structural, geomorphological and engineering geological mapping were conducted to acquire
geological, tectonic, geomorphological and rock mechanic properties (rock mass strength) characteristics.
Rock mass strength is obtained from the Engineering geological map of Hossana map sheet (Yekoye et al., 2012)
and Dilla map sheet (Habtamu et al., 2012). The maps are prepared based on extensive and multiple types of field
data to classify the lithological units into ranks of strength class as Very Low, Low, Medium, High, Very High rock
mass strength units. These classifications are based on multiple criteria evaluation determined from field
documentation including intact rock strength, discontinuity conditions and degree of weathering. The intact rock
strength determination is made either by Schmidt hammers or testing of representative irregular samples under the
point load tester and the results normalized to standard size of sample as recommended by ISRM (1985) to $IS_{50}$
reference strength. The discontinuity condition is determined by considering the spacing, aperture and discontinuity
surface roughness and overall geometry. The degree of weathering on the other hand is determined qualitatively on
the bases of the criteria set out in British Standard (BS 5930, 1981) from various outcrops in the region.
The precipitation data were obtained from the national database that was set up by the Centre for Development and
Environment (CDE), University of Bern, Switzerland in the 1990's for all of Ethiopia. Since its beginning, the
dataset has been upgraded with additional information layers but the dataset released as version I on a single CD-
ROM, which has mean monthly precipitation data of the major settlement areas with information on the temporal
coverage of recorded years, has been used in this study (CDE, 1999). Precipitation point data (Centre for
Development and Environment, 1999) were averaged (annual, each month) and then the spatial distribution over the
areas of interest were interpolated using the Inverse Distance Weighted method (IDW). Nevertheless, the
precipitation seasonality index could not be calculated due to data inhomogeneities, where only some stations have a
recording period of more than 20 yrs., but often less than 5 yrs. In order to calculate a seasonality index, 30 yrs.
continuity is required. Therefore, precipitation seasonality was evaluated using standard deviation among particular
monthly precipitations and by wet (July + August) and dry season (December + January) differences. Monthly
averages of all available data were considered for calculations.
Aster DEM, SRTM3 and Landsat 7 ETM+ were used for morphotectonic analysis, the vegetation index (NDVI)
based on Modis (Terra Modis, USGS eMODIS Africa 10-Day Composite) and land use / land cover data available
from the U.S. Geological Survey (https://earthexplorer.usgs.gov/, 2018) were also evaluated. Modis scenes from
January (peak of dry season) and August (peak of wet season) 2016 were used for the vegetation assessment.





The main approach for the morphotectonic analysis followed that used by Dhont and Chorowicz (2006 and
references therein). The main aim was to use DEM imagery to interpret the largest neotectonic structures in the
central and southern MER regions. Single-directional and multi-directional shaded reliefs and an elevation coloured
ASTER DEM image (Fig. 3) was generated using ArcMap 10.6 (www.esri.com). This DEM constitutes the basis for
morphotectonic analysis at the regional scale. The faults mapped can be considered as the main neotectonic faults
because they have a prominent expression in the morphology. In some cases they form asymmetric ranges with one
side corresponding to breaks in slope or scarps; by the displacement of Pleistocene and Neogene lithological
boundaries; by the occurrence of straight lines of kilometres to several tens of kilometres in length. The images were
compared with field geological mapping data to distinguish the scarps formed by active faults from those formed by
differential erosion of contrasted lithology.
The emplacement of volcanoes, which are abundant in study area, can also be related to tectonic structures such as
tension fractures or open faults. Small volcanoes arranged along the straight lines or linear clusters of adjacent
volcanoes were also interpreted as linear structures. The result of the interpretation is called "linear indices" which
mostly represent active faults (normal and normal-oblique slip), but because of uncertainties in detailed lithology in
some areas and a lack of field verification in some cases, the "linear indices" may also represent prominent fracture
zones, in exceptional cases, also lithological boundaries. To avoid such uncertainties, an independent evaluation of
the geomorphology by numerical methods was carried out. For an evaluation of the main tectonic indications of the
CMER and SMER, morphotectonic analysis was carried out at a regional scale of 1:250 000 (presented in sections
4.1. and 4.4.), while case studies Mejo and Arba Minch were evaluated on a detailed scale of 1: 50 000 (chapter
199  4.5.).
In addition to a visual interpretation of linear indices, a quantitative technique - morphometry - was also employed
to analyse landforms in a quantitative manner. This technique uses numerical parameters such as slope, surface
curvature and convexity to extract morphological and hydrological objects (e.g., stream networks, landforms) from
DEM (Fisher et al., 2004; Pike, 2000; Wood, 1996).  Landforms and lithological units reflect also different
geotechnical properties (e.g. rock strength, degree of weathering) so they can be identified by these numerical
methods. Various studies have been carried out to link morphometry with land erosion, tectonics and diverse
geomorphological conditions and volcanic activity (Altin and Altin, 2011; Bolongaro-Crevenna et al., 2005; Ganas
et al., 2005; Kopačková et al., 2011; Rapprich et al., 2010). Morphometric maps were constructed utilizing Wood's
algorithm based on SRTM DEM data (30 m pixel resolution). First, the topographic slope and the maximum and
minimum convexity values were derived at a pixel by pixel basis. The variation in these parameters was quantified
for each pixel with respect to neighbouring pixels (in orthogonal directions). Secondly, based on a set of tolerance
rules, morphometric classes were defined for each pixel: ridge, channel, plane, peak, pit and pass (Wood, 1996).
Wood's algorithm allows the relief to be parametrized by setting different values for the tolerance of the topographic
slope and convexity. In this study the slope tolerance of 3.0 and convexity tolerance of 0.02 were used for the best
fit.
**4.   Results and interpretations**
The results of the regional study of morphotectonics, morphometric and field structural analysis, slope failure
mapping and a geostatistical evaluation of the relationships between tectonic, lithological and surface conditions,
and the occurrence of the landslides are presented here. Also, a more detailed evaluation is finally carried out taking
two case study sites at Mejo (on MER eastern shoulder) and Arba Minch (western MER escarpment) areas which
have a contrasting geological and climatic setting across the MER.
**4.1.    Morphotectonic and morphometric analysis**
Shaded relief maps, derived from DEMs with NW, N and NE illumination, and multidirectional shaded relief maps
were used as a base map for morphotectonic interpretation. After carrying out the first stage of a visual interpretation
of the lineaments, the second stage was carried out on the automated/numerical morphology base map, which helped
uncover some important lineaments with a not so prominent morphological expression. Based on a comparison with
geological maps, lineaments representing lithological boundaries, without evidence of faults, were removed during
the third stage. Thus, the interpreted lineaments mostly represent present-day active faults, fault zones, important
fracture zones and possibly also shear zones (if there are any) which are manifested in morphology. Moreover, older
faults with a prominent lithological contrast can be expressed in morphology. The interpretation was made on a


scale of 1:250 000, so only the lineaments considered to represent a main fault or other tectonic zones have been
mapped.




*Fig. 2. DEM (colour elevation map on multidirectional shaded relief) of the Dilla and Hossana areas with visually*
*interpreted linear indices and the distribution of their strikes in a rose diagram. The location of the Mejo (Fig. 9)*
*and Arba Minch (Fig. 11) detailed study areas are also shown (see section 4.5).*
A combination of a visual morphotectonic interpretation based on DEMs (Fig. 2) and an interpretation on
morphometric landforms (Fig. 3) was used to map lineaments. The study area is characterised by a predominance of
NNE-SSW oriented lineaments mostly representing the major normal faults of the rift valley. The central and
northern parts of the study area represent a relatively wider rift zone with extension spread over a larger area, while
the southern part is narrower with steeper topographic gradients and more prominent vertical displacements on the
faults. The subordinate population of lineaments, mostly perpendicular to the strike of the rift has E-W to WNW
trend showing also vertical displacement.



*Fig. 3. Morphotectonic analysis of the Dilla and Hossana areas based on morphometry. Linear indices show only lines, which are in accordance with both the visual interpretation of the DEM and the morphometry.*

### 4.2. Tectonics

The primary fabrics in rift-related volcanic deposits and lava flows are defined by the planar preferred orientation of rock-forming minerals, micro-vesicles or micro-crystals and elongated mineral grains, lithic fragments or stretched and welded pumice fragments. With the exception of the lateral parts of lava flows or volcanic centres, these planar





fabrics are predominantly flat-lying or dip gently to ~SSW or E. In addition, large amount of fault structures
associated to the ~NNE-SSW trending MERS dip predominantly steeply to ~ESE in the western part of the rift and
to ~WNW along its eastern margin. The main ~NNE-SSW trending faults also form a prominent escarpments and
other morphological features of the MER (Figs. 4 a, 5). These faults are associated with fault lineation (slickensides)
plunging steeply to moderately to ~SE (in the western escarpment) or to ~NW (in the eastern escarpment), both
bearing exclusively normal kinematic indicators (Fig. 6 a, b, c). Two subordinate sets of fault structures appear to be
synchronous with the main ~NNE-SSW faults are mostly perpendicular, WNW(W)-ESE(E) trending normal faults
with predominantly NNW plunging slickensides or steeply ~NNW dipping normal faults (Fig. 5a). Relatively
younger or newly reactivated ~NNW(N)-SSE(S) trending faults which are oblique by ~20-30° to the main fault
system were mapped mainly in the central part of the rift valley (Fig. 2, 5a). In addition, ~NNW – ESE, ~NE-SW
and ~WSW – ENE trending strike-slip faults with a gently prevailing right-lateral kinematic pattern were identified
across the studied area (Fig. 2,5b). In spatial context of large volcanic centres (e.g. Wobitcha, Duguna Fango and
Awassa Caldera; Fig. 2) the caldera-related ring faults were found having a curved asymmetric shape, mostly
parallel to the caldera rim. These faults predominantly dip steeply to moderately inward to the centre of the caldera.
Extensional joints occur in three distinct sets with a ~ N – S, NNE – SSW and E (WNW) – W (ESE) trend (Fig. 5c).

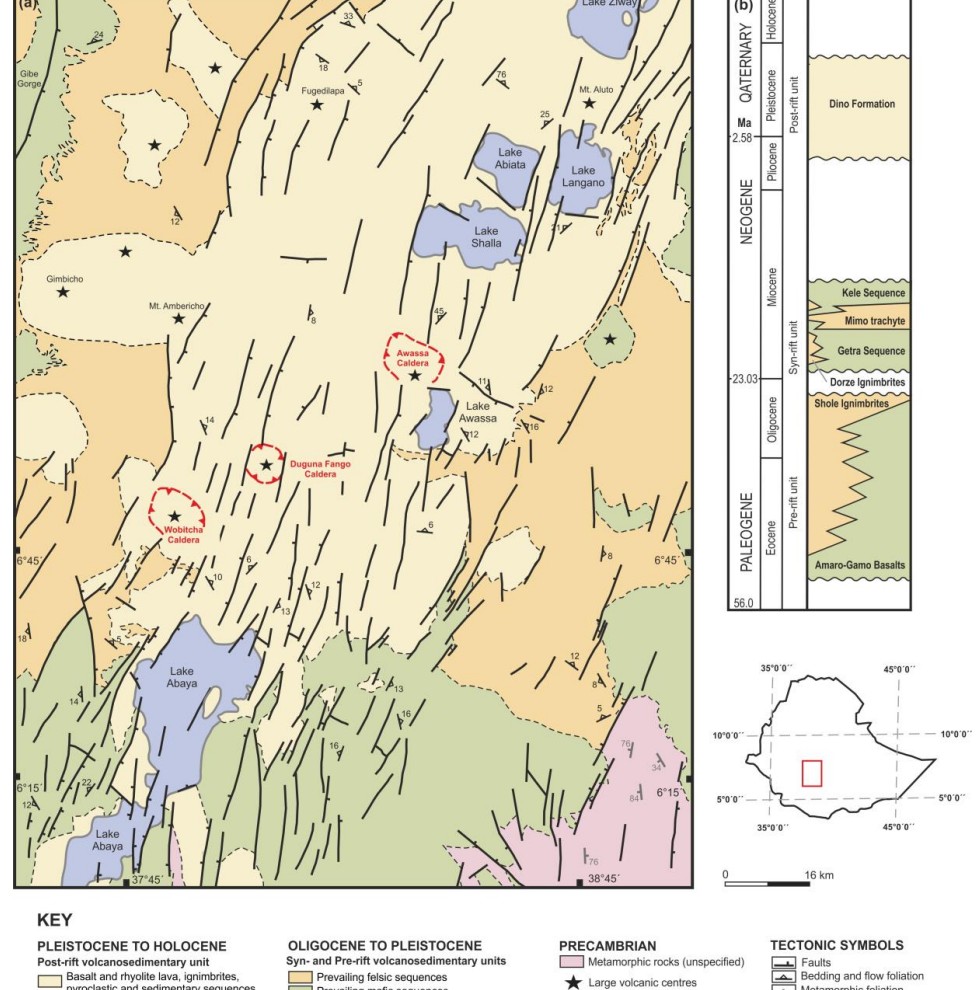


*Fig. 4. (a) Simplified geological map of the southern part of the Main Ethiopian Rift (Hossana and Dilla areas); (b)*
*Schematic stratigraphic chart of the Main Ethiopian Rift (Dilla and Hossana areas). Compiled using unpublished*
*geological maps 1:250 000 Geological Survey of Ethiopia.*


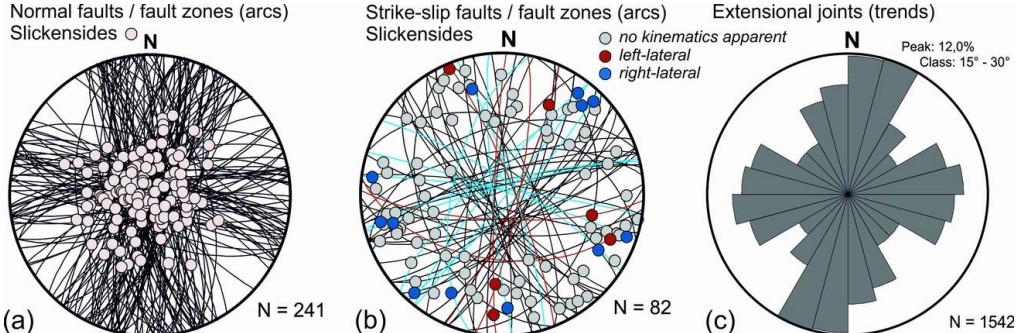


*Fig. 5. Field structural measurements of faults (equal area projection to lower hemisphere) and extensional joints*
*(rose diagram) from the southern part of the Main Ethiopian Rift (Hossana and Dilla areas).*

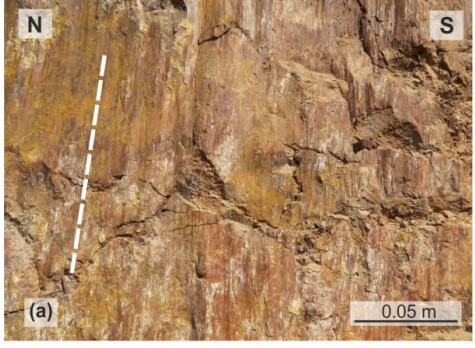
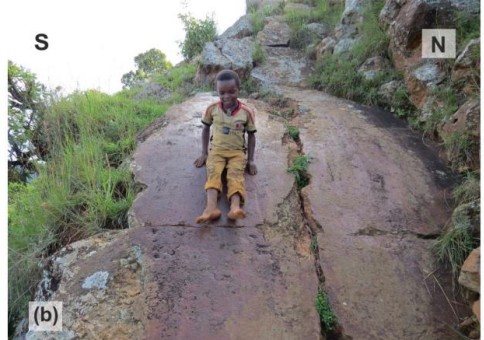

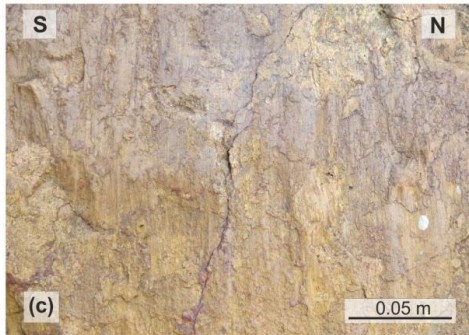
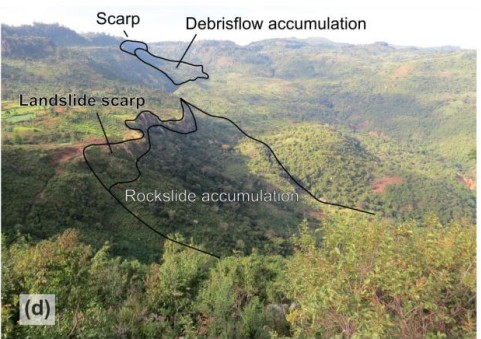


*Fig. 6. Field photographs (a) Steeply dipping, N – S oriented fault plane with steeply plunging slickensides and*
*normal kinematic indicators (west of Dilla Town, eastern rift escarpment). (b) ESE moderately dipping normal fault,*
*parallel with the main NNE-SSW trending western rift escarpment (Ocholo Village, north of Arba Minch). (c)*
*Steeply dipping, N – S oriented fault plane with steeply plunging slickensides and normal kinematic indicators (Mejo*
*Plateau, ca. 60 km east of the main rift valley). (d) Rockslide and debris flow on normal fault slope north of Arba*
*Minch.*

### 4.3.    Slope instabilities
Active extensional tectonics and the intense volcanism associated with the East African Rift System (e.g. Agostini et
al., 2011; Chorowicz, 2005) represent one of the main reasons for frequent hazardous geological phenomena in the
Main Ethiopian Rift (MER). Characteristic rift-related morphology, seasonal climatic conditions and inappropriate
human interference in the landscape create suitable conditions for hazardous geological processes. Endogenous risk
factors such as earthquakes, volcanism and post-volcanic phenomena are closely related with tectonics in this area.



The geomorphology is highly variable across the MER and is mainly the result of volcanic and tectonic events with
the associated erosional and depositional processes (Billi, 2015). The principal feature of the MER is the graben
bounded by normal faults. The drainage network is largely controlled by tectonic activity and lithological variation.
Parts of grabens form endorheic depressions filled by temporal lakes. The area is climatically highly variable; the
average amounts of annual rainfall vary from 500 in the Gibe and Omo Gorges to 2,600 mm on the escarpments and
the adjacent highlands. The mean annual temperature is about 20°C.
Slope failures, erosion, floods and the occurrence of ground fissures are the most common geological hazard
investigated in the Hossana and Dilla areas. Landslides, debris flows and rockfalls represent common exogenous
hazards distributed mainly on the fault scarps (Fig. 2 and 7 a). The subsidence of the rift floor and consequent uplift
of the highland lead to isostatic disequilibrium resulting in intensive head-ward erosion and slope processes. Most of
the slope instabilities represent deep seated complex fossil failures (Fig. 7 b) that host reactivated smaller landslides
and debris flows which are triggered by adverse anthropogenic practices (road construction, deforestation,
overgrazing) or river undercutting (fig. 7 e, f).
Rare lateral spread, with typical horst and graben features at the head, and many secondary shear surfaces have been
encountered in the complex un-welded ignimbrites and unconsolidated pyroclastic deposits with horizons of
paleosoils following the slip zone of this landslide (fig. 7 c). Topographic depressions with a higher degree of
saturation are often noted to have the long run effect of triggering landslides and debris flow on the slopes below
them (fig. 7 d, f).


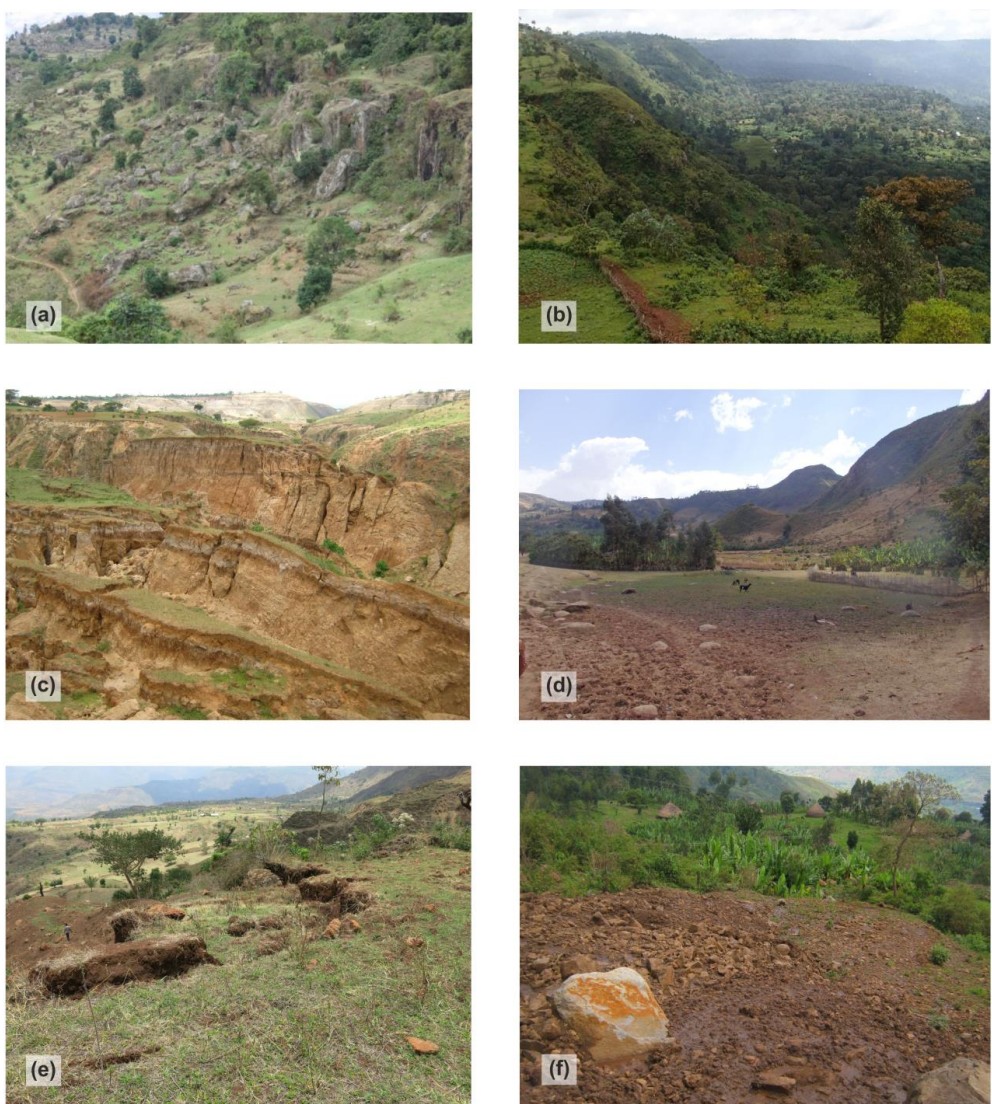

*Fig. 7. Field photographs of various types of geohazards in MER – Hossana and Dilla areas. (a) Toppling and*
*subsequent rock fall of welded ignimbrites in the crown of a deep-seated landslide situated close to a fault scarp in*
*the western highland area (Dilla area; NW of Arba Minch town). (b) Large landslide in Dilla area (5 km SW of*
*Mejo town). (c) Tilted blocks of deep-seated landslide southwest to Awassa. (d) Undrained depression in the deep-*
*seated fossil landslide east of Dilla Town. (e) Tension cracks in the crown of a shallow landslide reactivated by road*
*construction, west of Arba Minch. (f) Recent debris flow accumulation below road construction in the landslide area*
*west of Mejo.*

### 4.4.    Statistical analysis
Statistical analysis was carried out to better understand the influence of various surface processes and conditions
(precipitation, vegetation, slope, land cover) and geological parameters (rock mass strength, proximity of faults,
lineaments) on the formation of landslides and rockfalls. However, anthropogenic factors could not be evaluated
statistically because the relevant data are not available.

### 4.4.1.  Descriptive statistics

For the purposes of descriptive statistics, Rock Mass Strength (RMS) was coded as follows: Very High RMS = 7, High RMS = 6, Medium RMS = 5, Low RMS = 4, Very Low RMS = 3, Soils = 2, Lacustrine deposits = 1. A significant correlation between RMS and slope and most precipitation parameters was found (see Table 1). More wet and seasonal areas occur on steeper slopes formed by stronger (less weathered) rocks. Most of the steep slopes in the study area are active normal fault escarpments. Another interesting statistically significant correlation is shown by Slope and most of the precipitation parameters and the vegetation index (NDVI) of the dry period. Steeper slopes and higher altitudes are probably attracting clouds and precipitation, while flat lowlands allow clouds to pass by without precipitation. Significant correlations can also be found within various precipitation parameters, within selected vegetation parameters and also between these two groups (precipitation and vegetation), which was supposed. No significant correlation was found between the proximity of faults and lineaments (expressed by faults and lineaments density) and other parameters. It seems to be an independent variable very suitable for geostatistical evaluation.

| | RMS | Slope | Precipitation | | | | | NDVI | | | Faults and lineaments density |
| --- | --- | --- | --- | --- | --- | --- | --- | --- | --- | --- | --- |
| | | | Annual | Dry period | Wet period | Seasonality | Wet-dry period | Wet period | Dry period | Wet-dry period | |
| RMS | 1.00 | **0.44** | **0.49** | 0.17 | **0.43** | **0.58** | **0.39** | 0.10 | 0.07 | -0.01 | 0.13 |
| Slope | 0.44 | 1.00 | **0.37** | 0.11 | **0.25** | **0.37** | **0.22** | 0.16 | **0.24** | -0.12 | -0.11 |
| Precipitation annual | 0.49 | 0.37 | 1.00 | **0.61** | **0.47** | **0.73** | **0.35** | **0.28** | **0.37** | -0.16 | -0.14 |
| Precipitation dry period | 0.17 | 0.11 | 0.61 | 1.00 | -0.11 | -0.01 | **-0.27** | 0.14 | **0.41** | **-0.29** | -0.18 |
| Precipitation wet period | 0.43 | 0.25 | 0.47 | -0.11 | 1.00 | **0.80** | **0.99** | 0.15 | **-0.39** | **0.44** | 0.06 |
| Precipitation seasonality | 0.58 | 0.37 | 0.73 | -0.01 | 0.80 | 1.00 | **0.77** | **0.20** | 0.06 | 0.07 | 0.03 |
| Precipitation wet-dry period | 0.39 | 0.22 | 0.35 | -0.27 | 0.99 | 0.77 | 1.00 | 0.12 | **-0.44** | **0.47** | 0.09 |
| NDVI wet period | 0.10 | 0.16 | 0.28 | 0.14 | 0.15 | 0.20 | 0.12 | 1.00 | 0.16 | **0.46** | -0.05 |
| NDVI dry period | 0.07 | 0.24 | 0.37 | 0.41 | -0.39 | 0.06 | -0.44 | 0.16 | 1.00 | **-0.80** | -0.10 |
| NDVI wet-dry period | -0.01 | -0.12 | -0.16 | -0.29 | 0.44 | 0.07 | 0.47 | 0.46 | -0.80 | 1.00 | 0.06 |
| Faults and lineaments density | 0.13 | -0.11 | -0.14 | -0.18 | 0.06 | 0.03 | 0.09 | -0.05 | -0.10 | 0.06 | 1.00 |

*Table 1. Correlation matrix of the selected factors controlling distribution of geohazards in the MER area. Number of samples 153, critical value for correlation coefficient (R) at the 95 % significance level is 0.195. A statistically significant (95 %) R is in bold.*

### 4.4.2.  Geostatistics

The mean values of various geological, tectonic, climatic, vegetation and land use factors were calculated for each landslide polygon area. The normalized difference vegetation index (NDVI) is adopted from MODIS images of 2016 while density of lineaments is expressed as *[E+06]. The Kernel Density tool of the Spatial Analyst Tools/Density (ArcGIS 10.6) was used for evaluating the faults and lineaments density in MER on a scale of 1:250 000 (see Table 2). Proximity to tectonic features is expressed in terms of the percentage area of a particular geohazard within a particular buffer zone (500 m and 1 km buffer).

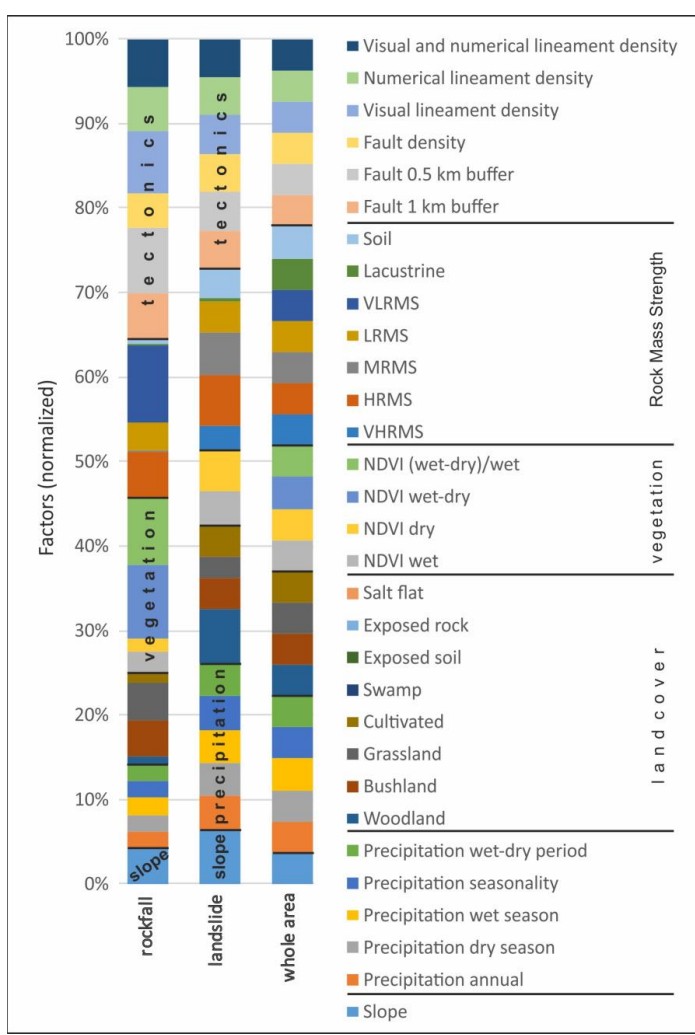

357

Fig. 8. Plot of mean values of particular factors occurring across landslides and rockfalls polygons normalized to the mean value for the whole area. Diagram shows the relative importance of each factor in comparison with the whole set of factors.

361

Most landslides and rockfalls form on steeper slopes close to faults and in areas with higher lineament density. Rockfalls are formed on steeper slopes than landslides (Table 2) but slope factor has higher importance for the formation of landslides (in comparison to other factors, see Fig. 8). Rockfalls typically occur on areas receiving lower precipitation. Most of them occupy areas with grassland and, to a lesser extent, also on cultivated land and bush land cover. Higher vegetation seasonality is also found to coincide well with rockfall occurrences. A low, very low and high rock mass strength class probably influence the occurrence of rockfalls (see Table 2 and Fig. 8). While landslides are formed in areas with higher precipitation and higher precipitation seasonality. Woodland, bushland, grassland and cultivated areas with higher vegetation density and low vegetation seasonality are found to have an affinity with landslide occurrences. All range of rock mass strength classes (low, medium and high) occur in areas of landslides.

372



| geohazard\factor | Slope [degree] | Precipitation | | | P. seasonality | | Vegetation | | V. seasonality | | Rock Mass Strength | | | | | | | Tectonics | | Lineaments density | | | | Landuse | | | | | | |
|---|---|---|---|---|---|---|---|---|---|---|---|---|---|---|---|---|---|---|---|---|---|---|---|---|---|---|---|---|---|---|
| | | annual [mm] | Dec+Jan (Dry) [mm] | Jul+Aug (Wet) [mm] | monthly 1σ | wet-dry [mm] | NDVI wet (Aug) | NDVI dry (Jan) | NDVI Aug-Jan | (Aug-Jan)/Aug [%] | VHRMS [%] | HRMS [%] | MRMS [%] | LRMS [%] | VLRMS [%] | Lacustrine [%] | Soil [%] | within 1 km buffer | within 0.5 km buffer | faults | visual | numerical | vis and num | woodland [%] | bushland [%] | grassland [%] | cultivated [%] | swamp [%] | exposed soil [%] | water [%] |
| rockfall | **17.2** | *1041* | 44 | 312 | 54 | 268 | **5412** | *3149* | **2263** | 42 | 0 | 27 | 3 | **40** | 25 | 1 | 3 | **88** | **66** | **155** | **341** | **227** | **227** | 8 | 18 | **48** | 21 | 1 | 0 | 4 |
| landslide | **15.6** | **1248** | 51 | **351** | **66** | **300** | 5296 | **5510** | *-214* | *-4* | 4 | 18 | **38** | 26 | 0 | 1 | 12 | 43 | 24 | 97 | 131 | 111 | 108 | **38** | 9 | 16 | **37** | 0 | 0 | 0 |
| whole area | 9.0 | 1172 | 48 | 333 | 61 | 285 | 4868 | 4297 | 571 | 12 | 5 | 11 | 28 | 26 | 6 | 11 | 13 | 36 | 19 | 82 | 103 | 95 | 88 | 22 | 9 | 24 | 36 | 1 | 1 | 6 |

*Table 2. Mean values for each geohazard polygon area compared to the whole area of Hossana and Dilla. NDVI calculated from Modis images 2016, lineaments density is \*[E+06]. The proximity of tectonics is expressed in the percentage area of the particular geohazard within the buffer. **Bold underline** - highly above average; **bold** - above average; italics - below average.*

### 4.5.    Case studies – Mejo and Arba Minch areas

Two areas with contrasting lithological, tectonic, climatic and vegetation settings and a similar size and morphology of landslides and rockfalls were selected for a detailed study. The study areas correspond with 1:50 000 map sheets (for location see Fig. 2).

#### 4.5.1.    Mejo Site

The Mejo study area is located 60 km east of the main rift valley on the upland plateau of the south-eastern flank of the MER. The Gambelto and Genale rivers drain the area southeast to Somalia form a typical morphology with deeply incised N-S trending valleys in the central part and volcanic plateaus along the south-western and eastern margin (Fig. 9). These volcanic plateaus attain an elevation slightly above 2000 m asl at east and around 2,100 m asl at south-west. Neoproterozoic medium-grade metamorphic rocks crop out mainly in the deeper part of the valleys below the altitude of ca 1900 m and the deepest parts reach below 1000 m asl. Thus, the area has a prominent topography with an altitude difference of more than 1000 m; the average slope in the area is more than 14 degrees. The overlaying volcanic deposits are of Eocene to Pleistocene age (Verner et al., 2018a; Verner et al., 2018b). The local climate is humid, the annual precipitation is  ~1,200 mm to ~1,550 mm (average 1393 mm) and highly seasonal usually with two peaks corresponding to April-May and August-October with more than 125 mm monthly average rainfall, while the rest of the months have a monthly average rainfall of slightly more than 40 mm. The difference between the average wet (July + August) and dry season (December + January) is 310 mm (CDE, 1999). Vegetation cover is dense (NDVI values almost double comparing the Arba Minch area) and moderately seasonal (see Table 3). Due to intense weathering the area is dominated by rocks with low and medium mass strengths. The dominant land cover is woodland and bushland, cultivated areas form up to 25 % of the area.

The area is formed by two units: (i) Metamorphic basement consisting of foliated biotite orthogneiss with minor lenses of amphibolites outcropped in the lower parts of the slope and the bottom of valleys. The orthogneiss is moderately to strongly weathered, the lenses of amphibolites have higher intact strength with a lower degree of weathering. The foliation of metamorphic rocks is often oriented downslope, parallel with the topography of the instable slopes. (ii) The volcanic complex overlying the metamorphic basement is formed by a roughly 500 metre thick succession of basalt and trachybasalt massive lava flows and intercalations of palaeosols, fine basaltic scoria layers and epiclastic deposits up to 2 m thick. The lava flows are moderately to strongly weathered with high fissured permeability, the pyroclastic layers, paleosols and strongly weathered horizons with high content of clay minerals may form semi-horizontal barriers for water movement resulting in higher plasticity and a reduction of permeability (Verner et al., 2018a; Verner et al., 2018b).

Most of the fault structures were identified in the complex of metamorphic rocks, without evidence of young reactivation. The youngest faults and fault zones belonging to the East African Rift System are rare and have no significant effect on the overall tectonic pattern of the area. These minor faults dip steeply to ~E or ~W, bearing well-developed steeply plunging slickensides and normal kinematics. The minor subordinate set of normal faults have a ~ W (WNW) to E (ESE) trend. The fault displacement is relatively low across the area, reaching a maximum of 100 metres in the vertical section (Verner et al., 2018a; Verner et al., 2018b). The prominent morphology, with up to 1000 m deeply incised valleys, is made almost solely by erosion caused by Neogene uplift.

A large and deep-seated complex landslide area occurs in the slope of the eastern banks of the Gambelto Valley. The landslide areas vary in length from several hundred metres to 4 kilometres, with a width of up to 2 kilometres (see Fig. 9). The landslide complexes are characterized by amphitheatre (horse-shoe)-shaped edges of the main scarps and reach up to 200 metres high, and 50 to 100 metre high minor scarps. Commonly, tilted blocks, endorheic



depressions and a number of springs have also been noted in the landslide zone. Reactivated parts are characterized
by small-scale (tens to hundreds of metres) and shallow-seated debris flows, slumps and rock-falls accompanied by
the subsidence of surface, cracks or curved tree trunks, which were observed close to the new road construction.
Most landslides are fossil and inactive. The preservation of colluvial deposits is limited, while in the depressed
domain and the arched accumulation area of the landslide they are covered by boulders and blocks. The morphology
of the main and minor scarps is relatively sharp and the accumulation zone is strongly modified by erosional
processes with a smooth and undulating topography, an absence of a hummocky landscape and traverse ridges. Most
of the reactivated parts are represented by small-scale and shallow-seated failures triggered by the poor design of
local road construction.

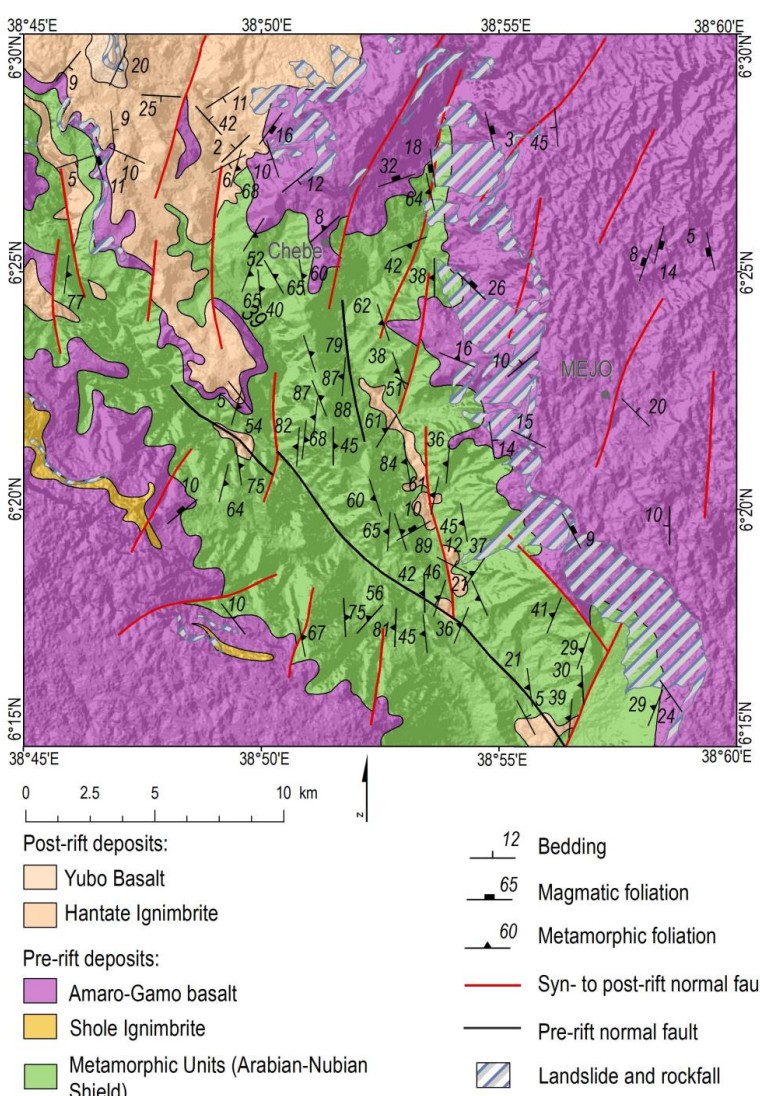

*Fig. 9. Geological and tectonic map of the Mejo Site with landslides and rockfalls indicated. For location, see Fig.*
*2.*




The mean values of the same factors as for the Hossana and Dilla areas (see section 4.4.2) were also calculated for
each landslide and rockfall polygon area in the case of the Mejo site. The same calculations and symbology as in
Table 2 was used for most parameters, but faults and lineaments data were adopted from more detailed studies at a
scale of 1:50 000 (Verner et al., 2018a; Verner et al., 2018b; Verner et al., 2018c; Verner et al., 2018d) and the faults
and lineaments density is calculated by a Line Density tool (ArcGIS 10.6. Spatial Analyst Tools) and expressed as
*[E+02]. Here the landslides and debris-flows are situated in areas with much higher slopes, compared to the overall
study area (see Fig. 10 and Table 3). They are also formed in areas with a higher vegetation density and medium and
low RMS. Landslide and debris-flow areas have a much higher density of lineaments. They are also dominantly
vegetated by woodlands, cultivated areas are a minor land cover.

| geohazard\factor | | Slope [degree] | Precipitation | | | P. seasonality | | Vegetation | | V. seasonality | | Rock Mass Strength | | | | | Tectonics | | | | Landuse | | | | |
|---|---|---|---|---|---|---|---|---|---|---|---|---|---|---|---|---|---|---|---|---|---|---|---|---|---|
| | | | annual [mm] | Dec+Jan (Dry) [mm] | Jul+Aug (Wet) [mm] | monthly 1σ | wet-dry [mm] | NDVI wet (Aug) | NDVI dry (Jan) | NDVI Aug-Jan | (Aug-Jan)/Aug [%] | VHRMS [%] | HRMS [%] | MRMS [%] | LRMS [%] | Lacustrine [%] | 1 km buffer [%] | 0.5 km buffer [%] | faults density | lineaments density | woodland [%] | bushland [%] | grassland [%] | cultivated [%] | water [%] |
| Mejo | landslide and debris-flow | 17.6 | 1335 | 46 | 346 | 75 | 300 | 6303 | 7278 | -975 | -0.15 | 2.06 | | 31.7 | 60.8 | 5.4 | 50.9 | 27 | 33.8 | 58 | 72 | 3 | | 26 | |
| | whole area | 14.2 | 1393 | 47 | 357 | 78 | 310 | 5548 | 6421 | -874 | -0.16 | 7.89 | | 28.3 | 41.9 | 22 | 61.5 | 36 | 33.6 | 34 | 53 | 19 | 3.1 | 24.8 | |
| Arba Minch | landslide and rockfall | 14.9 | 1070 | 60 | 188 | 45 | 128 | 5361 | 6412 | -1051 | -0.20 | | | 42.7 | 56.7 | 0.6 | 97.1 | 68 | 67.0 | 78 | | 30 | | 70 | |
| | whole area | 9.8 | 1068 | 59 | 189 | 46 | 130 | 3051 | 3909 | -858 | -0.28 | | 3.01 | 21.2 | 49.5 | 26 | 68.8 | 44 | 43.6 | 51 | 1.14 | 19.2 | | 51.2 | 28.4 |

*Table 3. Mean values for each geohazard polygon area compared to the overall area of Mejo and Arba Minch*
*respectively.*

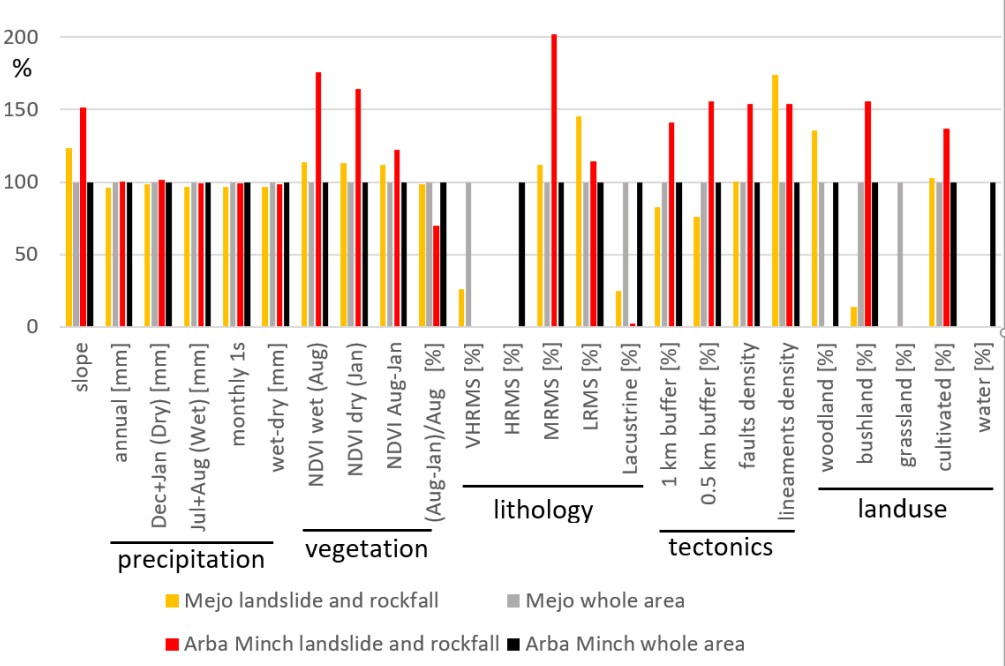

*Fig. 10. Plot of mean values of particular factors occurring across merged polygons of landslides and rockfalls*
*normalized to the mean value for the overall area. Mejo and Arba Minch sites evaluated separately.*

### 4.5.2. Arba Minch Site

The Arba Minch study area is located directly in the main rift valley on the western normal fault escarpment. The
total displacement of the syn- and post-rift normal faults is more than 1500 metres. The average slope in the area is
less than 10 degrees because a large part of the area is covered by Abaya Lake (see Fig. 11). The area is less humid,
compared to Mejo, with an average annual precipitation of 1068 mm and precipitation is moderately seasonal, the
difference between the wet and dry season is 130 mm. But significant variations in precipitation have been recorded



in apical parts of mountain ridges, such as Chencha, attaining, on average, an altitude of 2,700 m asl with 1,390 mm
of rainfall, whereas in the low-lying plains with an average elevation of about 1,200 m asl around the city of Arba
Minch the precipitation fluctuates around 780 mm (CDE, 1999). Vegetation cover is moderate (NDVI values almost
half of Arba Minch area) and moderately seasonal (see Table 3). Rocks with low and medium mass strengths and
lacustrine deposits dominate the area. The dominant land cover type is cultivated areas (form up to 51 %), bushland
and water surface are also abundant types. The area is formed by lower Eocene to Pleistocene volcanic and
volcaniclastic rocks, which are a product of episodic eruptions. They mostly have a bimodal composition with
alternating basic volcanic rocks and acidic pyroclastic rock intercalations (Verner et al., 2018 c; Verner et al.,
2018d). The prevailing faults are mostly parallel to the axis of the MER forming the area's prominent morphological
features. These major normal faults dip steeply to ESE or SE, trending NNE–SSE. Moreover, subordinate normal
faults were identified, predominantly steeply inclined faults trending WNW–ESE, which are perpendicular to the
prevailing rift-parallel normal faults. Fault displacement is relatively high across the area, reaching a minimum of
1,000 metres forming prominent morphology with an altitude difference of up to 1,500 m between the plateau and
graben floor.
The slope failures are located in the western steep fault scarps separating the bottom of the rift valley with Abaya
Lake representing a local erosional base at an elevation of 1,200 m asl and the western highland with an undulating
landscape at an elevation of between 2,000 and 2,400 m asl. The scarps are often modified by deep-seated slope
failures. The lower parts of the slopes form moderately weathered basalts and trachybasalt with minor pyroclastic
fall layers of volcanic ash reaching up to 2 m in thickness and a reddish paleosol up to 30 cm thick. The ridges and
upper parts are formed of welded ignimbrites with minor rhyolitic ash fall deposits and paleosol horizons. Volcanic
rocks are variably affected by intense fracturing, jointing and mega tectonic fault systems. Basalts and trachybasalts
are with a higher degree of weathering, while the welded ignimbrites with common columnar jointing are more
resistant. The volcanic units have fissured permeability. Mainly the ignimbrites represent rocks with high
permeability, on the other hand the highly weathered basalt, the intercalation of fine grained pyroclastics and
paleosol horizons could form hydrogeological horizontal barriers because of the high content of clay minerals. Most
of the landslides are represented by deep-seated complex slope deformations including toppling, rock-fall, rockslide,
rotational landslides and debris flows. These slope failures appear to be currently stable; the morphology is modified
by subsequent exogenous processes as in the Mejo area. Only several small-scale active landslides triggered by river
erosion and human intervention were observed.

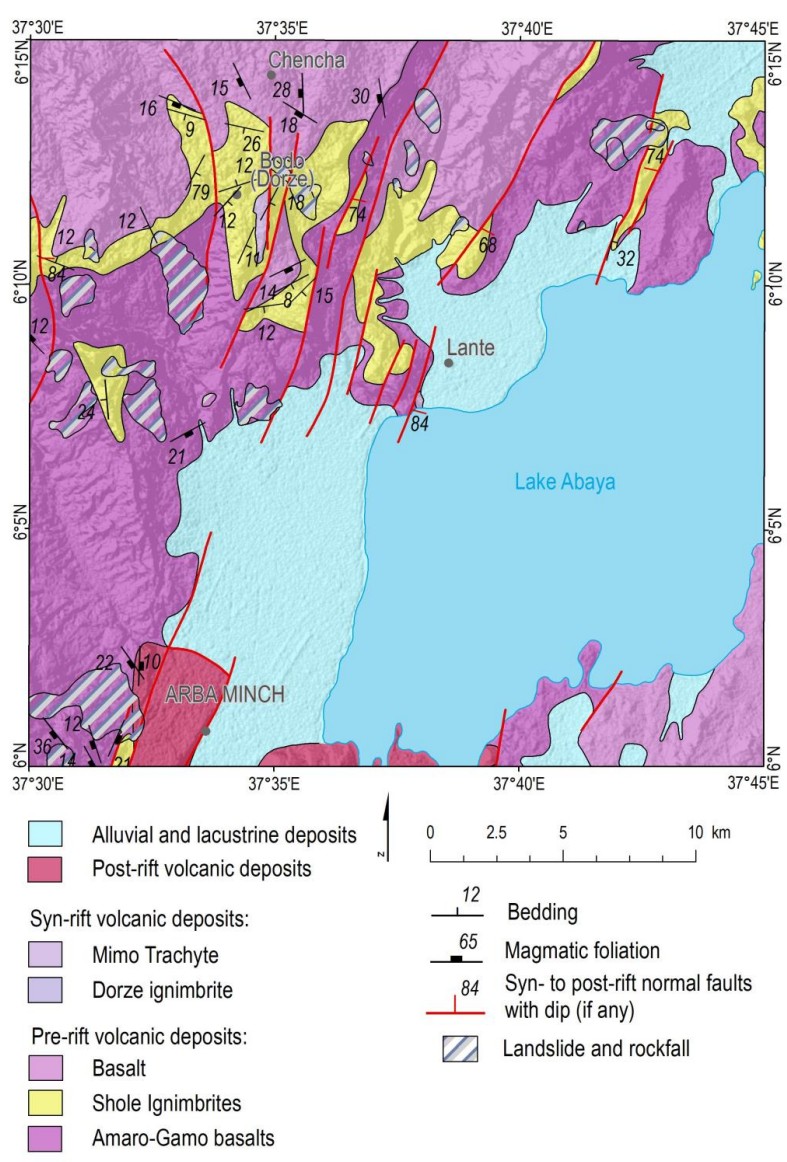

*Fig. 11. Geological and tectonic map of Arba Minch Site with landslides and rockfalls indicated. For location, see*
*Fig. 2.*
The mean values of the same factors as for the Mejo site were also calculated for each landslide and rockfall
polygon area at the Arba Minch site. Here the landslides and rockfalls are situated in areas with much higher slopes,
compared to the overall study area (see Fig. 10 and Table 3), there is a much higher density of faults and lineaments
close to faults. They are also formed in areas with much higher vegetation density and medium and low RMS.
Landslide and rockfall areas are also dominantly covered by cultivated areas with woodlands taking a minor role.





## 5. Discussion

### 5.1. Main Ethiopian Rift (Hossana and Dilla area)

The progressive changes of the paleo-stress regime during the active continental extension and faulting in the MERS (e.g. Corti et al. 2018; Zwaan and Schreurs, 2020) increase the tectonic anisotropy of rocks, slope instabilities along major and subordinate fault escarpments which have a pronounced effect on the genesis and formation of landslides. Several tectonic m,odels explain the kinematics and paleostress conditions of the regional extension / transtension from the beginning of the rifting (ca 12 Ma) to the present (for the review see Zwaan and Schreurs, 2020). Some models suppose continuous a NW – SE oriented extension (e.g. Chorowicz, 2005) in the early phase which later changed to its current E-W direction (Bonini et al., 2005; Wolfenden et al., 2004). Alternatively, other models also assume a permanent E – W to ESE – WNW oriented extension (e.g. Agostini et al., 2009; Erbello and Kidane, 2018).

Proximity to faults and lineaments have strong influence on the occurrence of rock falls and landslides in tectonically active areas worldwide (e.g. Chang et al, 2018; Kumar et al., 2019 and references therein). In the MER, both rockfalls and landslides typically occur on areas with steep slope, close to faults and with higher density of faults and lineaments. The latter parameter also reflects faults and fracture zone intersections and, according to geostatistic evaluation (Table 2), is more important for the formation of rockfalls than landslides. Rockfalls also show a much higher affinity to the proximity of faults.

Rockfalls occur in areas with lower precipitation, while for landslides high precipitation and high precipitation seasonality is typical. It correlates well with high vegetation density and low vegetation seasonality, which are found to have strong affinity with landslide occurrences. Thus, precipitation does not seem to be an important factor for rockfall formation but is important for landslides.

Rockfalls and landslides occur on areas with bushland, grassland and cultivated landcover. It leaves deforestation as one of the possible triggering factors. They also occur in areas with a wide range of rock mass strength classes (very low, low, medium and high) so lithology and intensity of weathering do not seem to be an important triggering factor.

In the large area of the MER the vast majority of slope instabilities is located on active normal fault escarpments (Fig. 12). This is a major natural triggering factor for rockfalls. While for landslides there is also the important influence of higher precipitation, precipitation seasonality and vegetation density and seasonality.

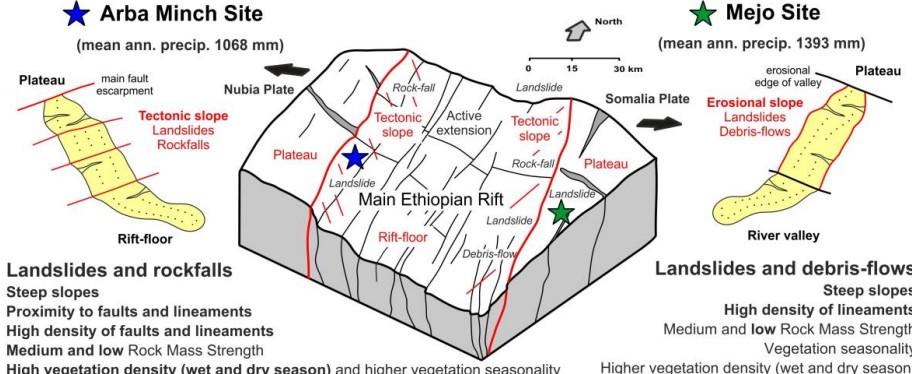

*Fig. 12. Sketch diagram summarising the main factors controlling the formation and distribution of particular slope failures in the MER and in the Arba Minch and Mejo study sites.*

### 5.2. Arba Minch case study

Slope instabilities, mostly landslides and rockfalls, here are situated in areas with much steeper slopes, a much higher density of faults and lineaments and close to major faults. The majority of the large-scale slope instabilities of this area is strongly associated with active tectonic morphological features characterized by straight fault scarps with triangular facets, large downthrown blocks, parallel sets of erosional valleys and asymmetrical ridges with SSW-


NNE trending. These features are associated with active normal faults having large displacements (total vertical
displacement of the western rift escarpment is more than 1500 m). Slope instabilities are also formed in areas with a
much higher vegetation density and medium and low RMS. Volcanic rocks are variably affected by intense
fracturing along faults, these zones are often altered, which lowers the slope stability of the rock environment.
Alteration is also enhanced by more intense water-rock interactions – most springs are located on fault zones (Arba
Minch means "Forty Springs"). Precipitation was not confirmed as an important factor.
The Arba Minch area is seismically active, according to the catalogue of earthquakes of the United States
Geological Survey (USGS) several earthquakes have been documented around Abaya Lake since 1973 with
magnitudes between 4 and 6 (USGS, Earthquake Hazards Program, 2017). This active tectonic is also documented
by young faults affecting Quaternary volcanic rocks and sediments outcropped around the town of Arba Minch
(Verner et al., 2018 c, d).
**5.3.    Mejo case study**
Landslides and debris-flows here are situated in areas with steep slopes. The geomorphology of the area is almost
unaffected by rift tectonics; evidence of young faulting as displacement of the Pleistocene and Holocene rocks,
straight fault scarps with triangular facets, has not been observed. The steep slopes are formed and strongly modified
by intensive head-ward erosion. The incision of the valley as a result of a lowered erosional level and highland uplift
could be the driving factor for the slope instability in the case of the Mejo area. Geomorphic proxies and the
thickness of flood basalts suggest that the more tectonically active south-eastern escarpment of the CMER and
SMER (where the Mejo site is situated) are experiencing a relatively higher rate of tectonic uplift compared to the
south-eastern escarpment of the northern MER and the Afar Depression (Xue et al., 2018; Sembroni et al., 2016).
This can also be noted from the Eocene-Oligomiocene basalts base (35 – 26 My) occurring in Arba Minch at an
elevation of around 1050 m asl compared to their occurrence at a much higher elevation in Mejo, at around 1900 m
asl (Verner et al., 2018a; Verner et al., 2018b; Verner et al., 2018c; Verner et al., 2018d).
Another factor causing the decrease of slope stability could be local lithological properties (dominance of medium
and low RMS characteristic for slope instabilities in the area): (i) frequent intercalations of palaeosols with a high
content of clay minerals and low permeability, (ii) a strongly weathered metamorphic basement with foliation often
concordant with the landscape forming a very weak lithological environment, which is favourable for slope
processes. No young volcanic features and products have been observed; the probability of earthquakes related to
volcanic eruptions is very low in the Mejo area, where the nearest earthquakes were recorded 60 km NW of the
study area.
**5.4.    Comparison of Arba Minch and Mejo sites**
Landslides at both sites are similar from the geomorphological point of view, i.e. old, stabilized, smoothed by
erosion. Young reactivations are very localized and mostly due to human activity. Both study areas have seasonal
humid climates with a prominent summer (mid June – mid September) rain season, but the Mejo study area, which
is situated 90 km east of Arba Minch, 60 km out of the main rift valley on the fast-uplifting plateau, is more humid.
In the Mejo area the mean annual rainfall is 30 % higher (1393 mm) compared to Arba Minch (1068 mm), most of
the precipitation difference falls in the rainy season, while during the dry months the precipitation at both localities
is comparable (Table 3).
Steep slopes associated with active faulting and hydrogeological conditions favouring rock alterations along these
zones are probably the main factors triggering the formation of slope instabilities in Arba Minch. In addition to these
factors, seismic events could also be speculated as one of the triggering factors.
The combination of a deeply weathered Proterozoic basement and steep slopes formed by intense head-ward
erosional processes due to rapid uplift could represent the main factors for creating favourable conditions for
landslide evolution in Mejo (Fig. 12). More intense precipitation may also contribute to slope instability.
**6.    Conclusions**
Active continental rifting has a distinct effect on the formation of landslides. The formation, superposition, and
polyphase reactivation of fault structures in the changing regional stress-field increase the tectonic anisotropy of
rocks and increase the risk of slope instabilities forming. The new structural data from the CMER and SMER
support a model of progressive change in the orientation of the regional extension from NW – SE to the recent
E(ENE) – W(WSW) direction driven by the African and Somalian plates moving apart with the presumable
contribution of the NNE(NE) – SSW(SE) extension controlled by the Arabic Plate.

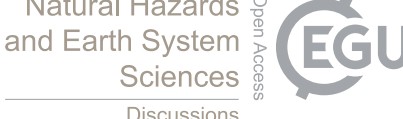

An evaluation at the regional scale of the central and southern MER demonstrates that slope instabilities, mainly landslides and rockfalls, occur on steep slopes, which are almost exclusively formed on active normal fault escarpments. Landslides are also importantly influenced by higher annual precipitation, higher precipitation seasonality and vegetation density and seasonality, while rockfalls have an affinity to vegetation seasonality only. Different geological, geomorphological, and climatic conditions can lead to formation of similar types of slope instabilities. A detailed study on active rift escarpment in the Arba Minch area revealed similar affinities as in the regional study of MER. Slope instabilities here are closely associated with steep, mostly faulted, slopes and a higher density of vegetation. Active tectonics and probably also seismicity are the main triggers. While the detailed study situated in the Mejo area on the uplifting Ethiopian Plateau 60 km east of the rift valley show that the occurrence of slope instabilities is strongly influenced by steep erosional slopes and deeply weathered Proterozoic metamorphic basement. Landslides here are often formed in areas densely fractured and with foliation concordant with topography. Rapid head-ward erosion, unfavourable lithological conditions and more intense precipitation and higher precipitation seasonality are the main triggers.

*Competing interests.* The authors declare that they have no conflict of interest.

*Acknowledgements*. The research was funded by the Czech Development Agency in the framework of development project No. 281226/2018-ČRA "Implementation of a Methodical Approach in Geological Sciences to Enhance the Quality of Doctoral Studies at the Addis Ababa University (Ethiopia)" (to K. Verner) and project No. 280614/2019-ČRA "Ensuring Sustainable Land Management in Selected Areas of Ethiopia on the Basis of Geoscientific Mapping" (to K. Verner). We thank our many colleagues from the Geological Survey of Ethiopia and Addis Ababa University (School of Earth Sciences) for their help in the acquisition, processing and interpretation of the data, especially to Aberash Mosisa and Wubayehu Dessalegn Sallile. Many thanks to Richard Withers for the English proof reading.

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
