# Peer review of "Main Ethiopian Rift landslides formed in contrasting geological 1"

_Natural Hazards and Earth System Sciences, 2020_

## Referee Comment (RC2)

[referee-annotated manuscript omitted]

---

## Author Comment (AC2)

[revised manuscript text omitted]

---

## Author Response (AR1)

Editor report

Editor comment: 1. Structure of the manuscript: there are intermixed topics where e.g. state-of-the-art portions are provided in the results section as well as others where discussion and results are mixed. Please revise all this throughout the whole document.

Author comment: I revised manuscript and put all literature research to introductory sections, results to Results section and all discussion to Discussion section.

EC: 2. Methodology is somewhat complex and unclear. Please rewrite and separate into separate sub-sections

AC: I rewrote, separating adopted data from our data and sub-sections also made.

EC: 3. The statistics and geostatistics parts are not well introduced and justified. Some of the results are interpreted and should be supported. Several statements are in contrast between different sections (see detailed reports by the reviewers). Please fix this.

AC: More introduction and comments to geostatistics is made. Also contrasting conclusions cleared, it was due to different scale, area and data used.

EC: 4. The discussion and the conclusion of your work, which seems to be mainly focusing on landslides, should be more detailed as to which concerns the slope processes themselves. Therefore, the manuscript could improve significantly if you would try to better link methods (specifically statistical) and discussion. For example, when you report possible correlations (or lack of them) among environmental factors and landslides, please try to highlight and support your statements with your provided results. Also on this, please see detailed comments in the reviewers' reports.

AC: Discussion and conclusion parts enhanced, more explanation added as you and anonymous reviewers suggested.

EC: Please also reply to all the remaining minor comments and requests of the referees, by providing a one-to-one reply on a separate document and by providing the modified manuscript.

AC: All comments replied here and later in this document and in separate pdf. Also, manuscript text with marked changes and final manuscript without markings are provided.

Thank you very much for your time and valuable comments.

I hope manuscript is improved now, if there is still anything unclear, I am happy to provide more explanation.

Karel Martinek
* * *
Anonymous Referee #1

RC: In section 2.2 There are several toponyms. It could be helpful to report them in a map.

AC: I added them to Fig 1

RC: Figure 1: please report from where the digital elevation model has been obtained and its resolution.

AC: We were using AsterDEM and SRTM3 (it is noted in Methods section) with resolution 30 m. I added this info also in fig. caption.

RC: Why did you chosen to remove all the boundaries between lithologies? Sometimes landslide slip surfaces can occur along boundaries between different lithologies.

AC: Yes, you are right, but in our case, the study area is highly weathered and rock properties and susceptibility to land sliding are more dependent on altitude, weather, and rock age rather than lithology, which is similar across the area – volcanics and volcaniclastics. I added more info in the text.

RC: Figure 3 in not so easy to read. There are similar colours in among passes, ridges and channels. Please try a better way to shoe your results.

AC: We edited colour symbology to better identify particular phenomena.

RC: Figure 8 is a niche way to show the statistic results but considering the whole area it seems that all the considered factors have the same importance. Please clarify this point.

AC: The diagram is showing the mean values of particular factors occurring across landslides and rockfalls polygons normalized to the mean value for the whole area. So because of normalization, the whole area values are constant. Maybe its confusing to show the column with whole area (it has no information value, it is here just for better understanding how rockfall and landslide values were calculated and normalized), I rather omitted it from diagram.

RC: Please add the Gambelto and Genale rivers in figure 9.

AC: Added.

RC: In the discussion section, please, try to justify why precipitation is not an important factor for rockfall formation.

AC: Its result of our statistical analysis, the precipitation values are virtually the same in rockfall areas as average for whole study area, no anomalies, no extremes as in the case of landslides (they have higher values of all precipitation parameters comparing to average whole area). Frankly speaking we are not 100 percent sure why, probably rockfalls are limited to very narrow areas along upper parts of fault escarpments, typically few hundreds of meters wide, its definitely major controlling factor. Also the resolution of climatic data is very poor, meteorological stations are often many tens of kilometres apart, so interpolation of precipitation grid can not have appropriate resolution. Also temporal resolution is poor, no one station has continuous data longer than 10 yrs, there are many gaps in dataset. I will add more explanation to the discussion.

Thank you very much for your time and valuable comments.

Karel Martinek

Anonymous Referee #2

RC: Chapters are written with different quality, possibly due to a large number of contributors, but at the end one person should do the redaction and avoid statement which are in contrary. The problem might be also in the scale. I suggest you work in a detailed scale, that you can be sure about the relation "cause - response".

AC: I edited text to make it more coherent, contrary statements were avoided, statements looking contrary due to different scales and data used were explained in more detail. Focus is on detailed scale now.

RC: in enclosed PDF

AC: in enclosed PDF

Thank you very much for your time and very valuable comments.

Karel Martinek

[revised manuscript text omitted]

---

## Author Response (AR2)

Thank you very much for acceptance of the manuscript.

I went through the text again, trying to find and fix inconsistencies and repetitions. I add few lines to the introduction of Discussion section to connect particular following subsections.

Also few inconsistencies between Discussion and Conclusion sections were edited to make it clear (predispositions and trigger factors are consistent now across the text including abstract).

Maps and captions were edited to fulfil journal standard. Source citation added to map captions.

Thorough English proof reading by native speaker with considerable experience with scientific texts Richard Withers were performed (in previous version in case of revised text it was not done).

best regards

Karel Martínek